# Can the Pathological Response in Patients with Locally Advanced Gastric Cancer Receiving Neoadjuvant Treatment Be Predicted by the CEA/Albumin and CRP/Albumin Ratios?

**DOI:** 10.3390/jcm13102984

**Published:** 2024-05-19

**Authors:** Ertugrul Bayram, Mehmet Mutlu Kidi, Yasemin Aydınalp Camadan, Sedat Biter, Sendag Yaslikaya, Tugba Toyran, Burak Mete, Ismail Oguz Kara, Berksoy Sahin

**Affiliations:** 1Department of Medical Oncology, Faculty of Medicine, Cukurova University, Adana 01250, Turkey; mehmetmutlukidi@gmail.com (M.M.K.); yaseminaydinalp23@gmail.com (Y.A.C.); sedatb23@hotmail.com (S.B.); drysendag@gmail.com (S.Y.); iokara@cu.edu.tr (I.O.K.); berksoys@hotmail.com (B.S.); 2Department of Medical Pathology, Faculty of Medicine, Cukurova University, Adana 01250, Turkey; tugbaolcan@hotmail.com; 3Department of Public Health, Faculty of Medicine, Cukurova University, Adana 01250, Turkey; burakmete2008@gmail.com

**Keywords:** pathological response, gastric cancer, neoadjuvant treatment

## Abstract

**Background**: The purposes of neoadjuvant chemotherapy are to tumor size to improve the tumor removal rate, extend survival, and prevent metastasis. In this study, the importance of CRP/albumin ratio and CEA/albumin ratio in the prediction of neoadjuvant treatment response in gastric cancer patients was evaluated. **Methods**: This study retrospectively included 135 gastric cancer patients who received neoadjuvant chemotherapy at Çukurova University Balcalı Hospital between January 2018 and December 2023. Preoperative CRP/albumin and CEA/albumin ratios were compared according to treatment response and multivariate logistic regression analysis was performed to determine the potential importance of these ratios in predicting pathological response. **Results**: The mean age of the 135 patients was 58.79 ± 10.83 (min = 26–max = 78). The CRP/albumin and CEA/albumin ratios were found to be significantly lower in patients who did not respond to neoadjuvant therapy. Each 1-unit increase in the CRP/albumin ratio was associated with a 1.16-fold decrease in the odds of pathological complete response to neoadjuvant therapy. Both CRP/albumin and CEA/albumin ratios were found to be significant in distinguishing neoadjuvant therapy response. The optimal cut-off value was 2.74 for the CRP/albumin ratio (sensitivity = 60%, specificity = 78.4%) and 1.40 for the CEA/albumin ratio (sensitivity = 74.2%, specificity = 67.6%). Values below these cut-off points favored neoadjuvant therapy response. Pathological complete response to neoadjuvant therapy was 4.75 times higher in patients with a CRP/albumin ratio below 2.74 and 5.14 times higher in patients with a CEA/albumin ratio below 1.40. **Conclusions**: Findings demonstrate that in patients with locally advanced gastric cancer receiving neoadjuvant treatment, CRP/Albumin and CEA/Albumin ratios are significant markers of pathological response.

## 1. Introduction

With almost a million cases yearly, gastric cancer is the fifth most prevalent cancer worldwide, yet it ranks third among the number of cancer-related fatalities and is more common in men [1,2]. Eastern Asia and Europe have the highest incidence, with an approximate 33% 5-year overall survival rate [3].

While there are other risk factors, including family history, smoking, and dietary habits, Helicobacter pylori bacteria has been linked to gastric cancer in the majority of cases [4]. Adenocarcinomas, which are categorized precisely based on anatomical site and histological type, account for almost 95% of gastric cancer cases. The two histological subtypes of gastric cancer, referred to as intestinal and diffuse as per the Lauren classification, differ in their molecular and clinical features [3,5].

Surgery is the mainstay of treatment for gastric cancer. Depending on the size, location, and other characteristics of the tumors, surgical procedure outcomes may differ. The success rate is high in patients with early diagnosis, but the majority of patients who undergo resection develop recurrence because symptoms appear late and approximately 65% of patients are diagnosed at locally advanced or metastatic stages. The preoperative phase must be refined to the point of establishing an accurate picture of the patient’s condition to initiate their treatment journey. For example, among the meticulously planned stages, determining the staging of patients and prioritizing treatments such as neoadjuvant chemotherapy are crucial steps. Therefore, various adjuvant and neoadjuvant treatment approaches, including chemotherapy, radiotherapy, chemoradiotherapy, targeted therapy, and immunotherapy, have been developed to improve the available treatment options and outcomes after surgery [3,6,7].

Neoadjuvant chemotherapy is used to increase tumor resection rates by reducing tumor volume, preventing metastasis, and prolonging survival, and some studies suggest that it may help reduce unnecessary complications after surgery [8]. FLOT—5-fluorouracil (5-FU), leucovorin, oxaliplatin, and docetaxel—has become the accepted neoadjuvant chemotherapy regimen based on survival comparisons [9].

Factors such as age, gender, tumor type and localization, general health status, stage of the disease, presence or absence of lymph node involvement, and nutritional status may affect treatment success. Inflammation may increase angiogenesis, metastasis, and genetic instability and is known to decrease response to chemotherapy [4,10].

When inflammatory conditions occur, the production of C-reactive protein (CRP), an acute-phase protein, is induced by several cytokines. Several studies have shown that extensively high CRP levels predict resection and prognosis. Likewise, low serum albumin has been linked to poor prognosis and mortality, and low CEA levels were found to be a risk factor for poor prognosis in cases of gastric cancer [11,12].

Studies in the field of clinical oncology have investigated the relationship between systemic inflammatory response parameters and many inflammatory index markers such as hemoglobin, albumin, lymphocyte, platelet score (HALP), platelet/lymphocyte ratio (PLR), prognostic nutrition index (PNI), and CRP/albumin ratio with prognosis [13].

The aim of this study was to investigate whether the CRP/albumin ratio and the CEA/albumin ratio are useful in predicting clinical and pathological responses in patients receiving neoadjuvant therapy.

## 2. Materials and Methods

### 2.1. Study Design and Patients

This study is a retrospective single-center investigation. The sample size was determined to be a minimum of 128, considering a power (1–β) of 80%, a type 1 error (α) of 0.05, and an effect size (d) of 0.5 for a two-tailed hypothesis test. Between 2018 and 2023, a total of 390 patient files were reviewed, and 135 patients meeting the inclusion/exclusion criteria for the 5-year period were included in the study (Figure 1).

### 2.2. Inclusion and Exclusion Criteria

The inclusion criteria for the study were patients over 18 years of age who had not undergone surgery before and who had been diagnosed with a new gastric cancer without metastasis.

The exclusion criteria for the study were: (1) patients below the age of 18 years, (2) patients with organ failure, (3) being unable to adjust to therapy (mental health issues, hypersensitivity to medications, etc.), and (4) patients not being suitable for chemotherapy treatment before surgery.

The following information about the patients was taken from electronic medical records: demographic data; laboratory data (such as serum albümin (g/dL), CRP levels (mg/L), complete blood count, renal function, liver function, and CEA test (ng/mL) results; and pathologic findings.

PNI was calculated using a formula that incorporates serum albumin level and total lymphocyte counts.

The formula for calculating PNI is as follows [14]:PNI = 10 × serum albumin concentration (g/dL) + 0.005 × lymphocyte count (number/mm^2^),

### 2.3. Treatments

A FLOT or FOLFOX regimen was used in neoadjuvant treatment. FLOT treatment was given every 2 weeks as Oxaliplatin 85 mg/m^2^ for 2 h on day 1, Docetaxel 50 mg/m^2^ for 1 h on day 1, Folinic acid (Leucovorin) 200 mg/m^2^ for 2 h on day 1, and Fluorouracil 2600 mg/m^2^ for 24 h on day 1 (administered intravenously) (Table 1). FOLFOX treatment was given once every 2 weeks as Leucovorin 200 mg/m^2^ for 2 h with concurrent Oxaliplatin 85 mg/m^2^, and 5-Fluorouracil (5-FU) 2400 mg/m^2^ by continuous infusion in the first 48 h.

### 2.4. Pathological Evaluation

Resected gastric tissues were fixed in formaldehyde solution and subsequently embedded in paraffin. Histological tissue sections were obtained and stained with hematoxylin and eosin (H&E).

Treatment response was assessed based on morphological changes observed in the H&E-stained sections. The following criteria were used for scoring [15,16]; absence of viable cancer cells (complete response, score 0), presence of individual tumor cells or rare small tumor cell clusters (Near complete response, score 1), presence of tumor regression with more than individual cells and small tumor cell clusters (partial response, score 2), absence of tumor regression (No response, score 3).

The prepared tissue sections were evaluated under optical microscopy (BX43; Olympus, Tokyo, Japan) at different magnifications to assess treatment response and tumor morphology. In some cases, immunohistochemical staining with pankeratin (AE1/AE3/PCK26) antibody was performed to enhance visualization of tumor cells.

### 2.5. Surgery Evaluation

Patients receiving perioperative chemotherapy were evaluated at Cukurova University General Surgery. Surgically, total gastrectomy and D2 dissection were performed on the patients. D2 dissection or lymphadenectomy is considered the standard in curative surgery for patients with gastric cancer. The material taken after surgery was sent for pathological evaluation [17].

### 2.6. Ethical Statement

The study was approved by the Çukurova University Clinical Research Ethics Committee (decision no: 35/2.04.2022) and was conducted only with volunteer participants in accordance with the principles of the Declaration of Helsinki. Participants were given detailed information and signed written informed consent forms.

### 2.7. Statistical Analysis

In this study, the SPSS 20 (IBM SPSS Statistics for Windows, Version 20.0. Armonk, NY, USA) program was used for data analysis. Data were presented as numbers, percentages, arithmetic means, standard deviations, and medians. Shapiro–Wilk test was used as a normal distribution test. In normally distributed data, non-parametric tests were used for non-parametric data. Student *t*-test, Mann–Whitney U test, Binary logistic regression analysis, and ROC analysis were used in the analyses. *p* < 0.05 was considered statistically significant.

## 3. Results

### 3.1. Patient Characteristics

The mean age of the 135 patients included in our study was 58.79 ± 10.83 years (min = 26–max = 78). In terms of sociodemographic characteristics, the study comprised 93 (68.9%) males and 42 (31.1%) females. Pathologically, 89 cases (65.9%) were diagnosed as adenocarcinoma, 41 (30.4%) as squamous cell carcinoma, and 5 (3.7%) as mucinous carcinoma. Regarding tumor localization, the majority were found in the intestinal region (117 cases, 86.7%), followed by the cardia (44 cases, 32.6%), antrum (39 cases, 28.9%), the esophago–gastric junction (15 cases, 11.1%), corpus (28 cases, 20.7%), and diffuse (8 cases, 5.9%). The most common tumor type was intestinal (86.7%), while 13.3% were of the diffuse type. Neoadjuvant chemotherapy was administered, with 99.3% of patients receiving the FLOT regimen and 0.7% receiving the FOLFOX regimen. Regarding pathological response, 30.4% showed a complete response, 42.2% showed a partial response, and 27.4% were non-responsive (Table 2, Figure 2).

### 3.2. Associations between İnflammatory İndex and Pathological Response to Neoadjuvant Treatment

When the PNI index, PLT/MPV ratio, CRP/albumin ratio, and CEA/albumin ratio were compared based on the results of pathological response to neoadjuvant treatment, it was found that the PNI index, CRP/albumin ratio, and CEA/albumin ratio were statistically significantly lower in patients who did not respond to treatment (Table 3).

### 3.3. Predictors of Pathological Response

Logistic regression analysis (Forward LR method) used to predict the pathological response (no response/response) to neoadjuvant treatment was significant (omnibus test *p* = 0.002). The dependent variable in the model was the pathological response (predicted: response) and the independent variables were the PNI index, PLT/MPV ratio, CRP/albumin ratio, and CEA/albumin ratio. The explanatory power of the model was 10.3%. The accuracy of the model was 71.4%. Among the variables included in the model, the CRP/albumin ratio made a significant contribution to the model, with each 1 unit increase in this ratio decreasing the probability of response 1.16 times (16%) (Table 4).

When the significances of the CRP/albumin and CEA/albumin ratios obtained before neoadjuvant treatment were analyzed to determine their power in predicting responses to treatment, it was found that the areas under the curve were significant, indicating that these ratios serve as diagnostic tests with moderate power (Figure 3, Table 5).

According to our results, the optimum cut-off value of the CRP/Albumin ratio was found to be 2.74 (sensitivity 60%, specificity 78.4%), and that of the CEA/Albumin ratio was found to be 1.40 (sensitivity = 74.2%, specificity = 67.6%). Values below these values favored responses to neoadjuvant treatment (Table 6).

According to the results of our study, logistic regression analysis of the cut-off values used to predict the pathological response (no response/response) to neoadjuvant treatment was significant (omnibus test *p* < 0.001). The dependent variable in the model was pathological response (predicted: response) and the independent variables were CRP/albumin ratio and CEA/albumin ratio. The explanatory power of the model was 31%; the accuracy of the model was 79.4%. Among the variables included in the model, each 1.55 unit increase in the CRP/albumin ratio in patients with a CRP/albumin ratio below 2.74 increased the probability of response 4.75 times, and each 1.63 unit increase in the CEA/albumin ratio increased the probability of response 5.14 times (Table 7).

## 4. Discussion

Gastric cancer ranks third in terms of cancer-related mortality and is one of the most common cancers worldwide, with about one million cases annually. The frequency of gastric cancer is steadily declining, the epidemiology of the disease has changed over time, and the best available diagnosis and treatment methods are effective in this instance [18]. In our present study, we investigated the relationship between pathological response and the CRP/albumin and CEA/albumin ratios in patients receiving neoadjuvant therapy for gastric cancer.

Although surgery is the mainstay of gastric cancer treatment, most patients develop recurrence after surgery. There is no effective treatment for recurrent gastric cancer; therefore, it is necessary to estimate the risk of recurrence in patients and to plan an adequate follow-up program in clinical practice with the application of adjuvant and neoadjuvant therapies. Clinical trials to identify subgroups with the same risk while analyzing the efficacy of different treatments are important to follow the prognosis [19]. Our study’s findings indicate that the CRP/albumin ratio and the CEA/albumin ratio were statistically significantly lower in individuals who did not respond to treatment and that they were diagnostic tools with moderate power for discriminating responsiveness to neoadjuvant treatment.

There is no standard protocol for choosing which patients should receive neoadjuvant chemotherapy; some medical professionals pursued enhanced therapy for patients with advanced stage, for whom a longer duration of preoperative chemotherapy was preferred, while others opted for patients with radiological or clinical responses [20].

Inflammation is associated with tumor growth and development by changing the cancer microenvironment [21]. The dietary status and metabolic needs of a patient have an effect on albumin levels. Low albumin has been associated with inflammation [22]. Likewise, CRP is an acute-phase protein that is produced in the liver and stimulated by cytokines that promote inflammation [23]. Few studies have investigated the use of preoperative parameters such as albumin levels to predict the risk of postoperative recurrence and complications in patients undergoing neoadjuvant therapy [24]. Yu et al. [25] found that low preoperative albumin levels were a risk factor (*p* = 0.033) for postoperative complications in patients undergoing curative gastrectomy after neoadjuvant chemotherapy, while Migita et al. [14] found a correlation between decreased serum albumin levels after neoadjuvant therapy and survival rate.

Research has demonstrated that a number of inflammatory mediators, including, chemokines, acute phase proteins, and cytokines, are crucial for the angiogenesis, invasion, and metastasis of cancer. Several inflammation markers have been used to assess the prognosis of gastric cancer, including mGPS, PLR, CRP/Albumin, and HALP [26,27].

In many studies evaluating patients with advanced gastric cancer who underwent surgery, it has been emphasized that the CRP/albumin level is an important factor in predicting prognosis and mortality [27,28]. In addition to previous studies, we observed that the CRP/albumin and CEA/albumin ratios can be used to track the efficacy of treatment and predict the pathological response of patients experiencing gastric cancer [29]. Since these patients are receiving neoadjuvant chemotherapy, which has a significant effect on the patients’ ultimate prognoses, it is critical to develop an intuitive and reliable approach for estimating the consequences of chemotherapy. It may help physicians in making clinical decisions by identifying the patient categories most likely to benefit from neoadjuvant chemotherapy for gastric cancer. The CRP/albumin and CEA/albumin cut-off values, respectively, are set at 2.74 and 1.40. The differences might be connected to the patients’ disparate backgrounds.

In our case, the condition was locally advanced. However, in cases of peritoneal carcinomatosis, which are evaluated using laparoscopy to establish peritoneal cancer index (PCI), bidirectional therapy can be implemented with the recent introduction of Pressurized Intraperitoneal Aerosol Chemotherapy (PIPAC). It is worth noting that even in such scenarios, achieving conversion therapy followed by surgical intervention is feasible. Furthermore, it is essential to acknowledge that the culmination of these therapeutic interventions often involves adjuvant treatment [30], which serves to complement the overall treatment approach, with the previously mentioned indices retaining their significance in monitoring and assessing treatment outcomes.

In order to prevent unnecessary surgery in gastric cancer patients and to identify new markers that predict pathological response for better therapy, it is important to evaluate the preoperative status of the patient, compare the blood values of the patients, and continuously monitor them.

There are limitations to our study. The small sample size in our analysis may help to explain this finding. Aside from this, our retrospective study conducted at a single center has certain limitations, including bias in information gathering and selection. Larger sample sizes and prospective multicenter investigations are required.

## 5. Conclusions

According to the results of this study, CEA/albumin and CRP/Albumin ratios may be important in predicting pathological response in patients with locally advanced gastric cancer receiving neoadjuvant therapy and surgery. Since our results are not robust, they should be interpreted with caution by clinicians, and the results of our study should be supported by larger multicentre studies.

## Figures and Tables

**Figure 1 jcm-13-02984-f001:**
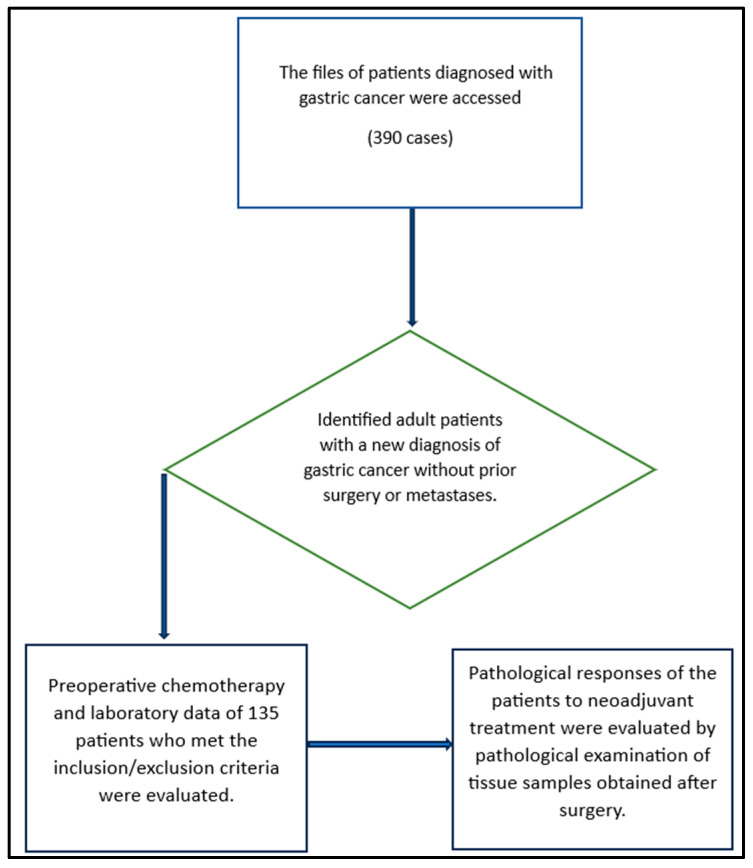
Flow chart for the selection of patients.

**Figure 2 jcm-13-02984-f002:**
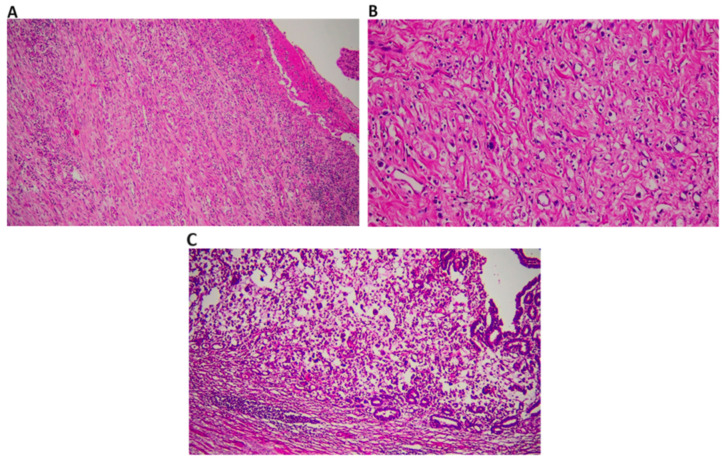
In complete response, the tumor has completely transformed into fibrosis, leaving no live tumor cells observable on the ulcerated surface of the gastric mucosa. We take care to sample the entire tumor bed macroscopically in these cases. If necessary, staining methods such as pankeratin are utilized in cases of suspicion ((**A**), H&E ×200). Only a few atypical cells are noticeable, standing singly on a fibrotic background. We interpret this phenomenon as a partial response ((**B**), H&E ×400). In Adenocarcinoma, no regression is observed in the tumor showing signet ring cell morphology after treatment. We evaluate this phenomenon as unresponsive ((**C**), H&E ×200).

**Figure 3 jcm-13-02984-f003:**
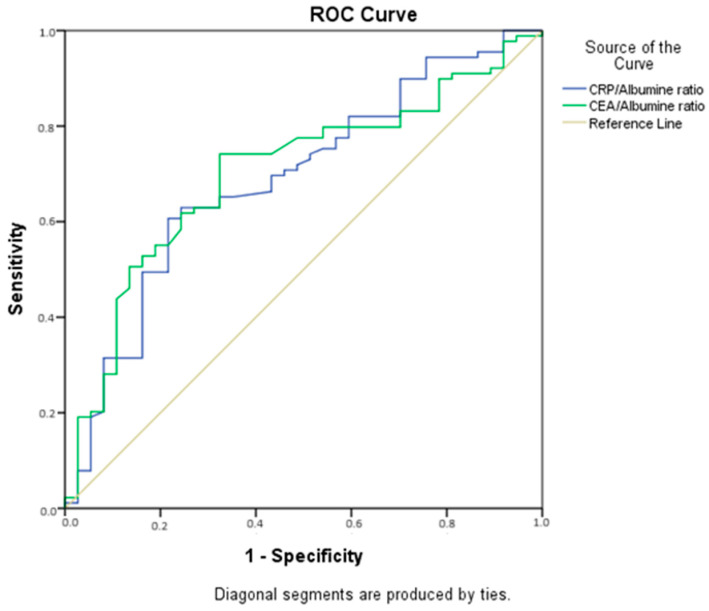
Areas under the curve for CRP/Albumin and CEA/Albumin ratios.

**Table 1 jcm-13-02984-t001:** Chemotherapy protocol.

FLOT Treatment (Every 2 Weeks)	FOLFOX Treatment (Every 2 Weeks)
Oxaliplatin 85 mg/m^2^ for 2 h on day 1 Docetaxel 50 mg/m^2^ for 1 h on day 1 Folinic acid (Leucovorin) 200 mg/m^2^ for 2 h on day 1 Fluorouracil 2600 mg/m^2^ for 24 h on day 1 (administered intravenously)	Leucovorin 200 mg/m^2^ for 2 h with concurrent Oxaliplatin 85 mg/m^2^ 5-Fluorouracil (5-FU) 2400 mg/m^2^ by continuous infusion in the first 48 h

**Table 2 jcm-13-02984-t002:** Sociodemographic characteristics.

	Mean ± SD (min–max)	
**Age**	58.79 ± 10.83	
**Preop CEA**	7.63 ± 14.40 (0.33–79)	
**Preop CRP**	13.39 ± 12.23 (1–60)	
**Preop albumin**	3.57 ± 0.54 (2.1–4.7)	
**Sex**	** *n* **	**%**
Male	93	68.9
Female	42	31.1
**Pathology**		
Adenocarcinoma	89	65.9
Signet ring	41	30.4
Mucinous	5	3.7
**Tumor location**		
Cardia	44	32.6
Corpus	28	20.7
Antrum	39	28.9
Diffuse	8	5.9
Esophagogastric junction	15	11.1
**Tumor type**		
Diffuse	18	13.3
Intestinal	117	86.7
**Neoadjuvant chemotherapy**		
FOLFOX	1	0.7
FLOT	134	99.3
**Pathological response**		
Complete response	41	30.4
Partial response	57	42.2
Unresponsive	37	27.4
**ECOG-PS**		
0	119	88.1
1	15	11.1
2	1	0.7
**T stage**		
T2	14	10.4
T3	70	51.9
T4a	44	32.6
T4b	7	5.1
**N stage**		
N0	7	5.2
N1	43	31.9
N2	42	31.1
N3a	40	29.6
N3b	3	2.2
**Total**	135	100.0

SD, standard deviation; Preop, preoperative; CRP, C-reactive protein; CEA, carcinoembryonic antigen; ECOG-PS, Eastern Cooperative Oncology Group performance score.

**Table 3 jcm-13-02984-t003:** Comparison of indices based on pathological response.

	Pathological Response				
	Responsive		Unresponsive		
	Mean ± SD	Median (IQR)	Mean ± SD	Median (IQR)	*p*
PNI Index	37.96 ± 3.83	38.00 (4.80)	34.28 ± 4.93	34.00 (6.5)	<0.001
PLT/MPV ratio	34,182.25 ± 15,235.52	30,602.40 (18,782.91)	31,372.23 ± 12,114.25	30,258.58 (17,482.81)	0.423
CRP/Albumin ratio	5.47 ± 4.71	3.84 (4.47)	3.12 ± 3.11	2.19 (3.03)	0.001
CEA/Albumin ratio	3.05 ± 4.71	1.77 (1.07)	1.73 ± 3.42	0.59 (1.20)	<0.001

SD, standard deviation; PNI, prognostic nutritional index; PLT, platelet; MPV, mean platelet volume; CRP, C-reactive protein; CEA, carcinoembryonic antigen.

**Table 4 jcm-13-02984-t004:** Logistic regression analysis for pathological response prediction.

	B	*p*	OR	95% CI for OR	
				Lower	Upper
CRP/Albumin ratio	−0.156	0.004	0.855	0.770	0.950
Constant	1.517	<0.001	4.557		

OR, odds ratio; CI, confidence interval; CRP, C-reactive protein.

**Table 5 jcm-13-02984-t005:** The areas under the curve.

Test Result Variable(s)	Area	Std. Error	*p*	95% CI	
				Lower Bound	Upper Bound
CRP/Albumin ratio	0.689	0.052	0.001	0.588	0.790
CEA/Albumin ratio	0.703	0.049	<0.001	0.606	0.799

CI, confidence interval; CRP, C-reactive protein; CEA, carcinoembryonic antigen.

**Table 6 jcm-13-02984-t006:** Validity results for CRP/Albumin ratio and CEA/Albumin ratio cut-off values.

Test Result Variables	Positive If Less than or Equal to	Sensitivity	Specificity	Youden Index	LR(+)	LR(−)
CRP/Albumin ratio	2.7402	0.607	0.784	0.391	2.81	0.50
CEA/Albumin ratio	1.4049	0.742	0.676	0.418	2.29	0.38

LR(+), likelihood ratio positive; LR(−), likelihood ratio negative; CRP, C-reactive protein; CEA, carcinoembryonic antigen.

**Table 7 jcm-13-02984-t007:** Logistic regression analysis for pathological response prediction.

	B	*p*	OR	95% CI for OR	
				Lower	Upper
CRP/Albumin ratio	1.559	0.001	4.75	1.85	12.16
CEA/Albumin ratio	1.637	<0.001	5.14	2.13	12.39
Constant	0.972	<0.001	2.64		

OR, odds ratio; CI, confidence interval; CRP, C-reactive protein; CEA, carcinoembryonic antigen.

## Data Availability

Data are available on request from the authors.

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
