# Peer review of "Can the Pathological Response in Patients with Locally Advanced Gastric Cancer Receiving Neoadjuvant Treatment Be Predicted by the CEA/Albumin and CRP/Albumin Ratios?"

_jcm, 2024, doi:10.3390/jcm13102984_

Round 1
Reviewer 1 Report
Comments and Suggestions for Authors
The Japanese study on the CRP/albumin ratio and the CEA/albumin ratio in predicting the response to neoadjuvant treatment in patients with gastric cancer is an intelligent reflection and aims to certify a unit of measurement on the progress of the therapies proposed to patients affected by this serious illness. For those dealing with gastric cancer there are important end points that ultimately lead to resective therapy. The preoperative phase must be refined to the point of having a precise photograph of the patient to begin a therapeutic path.
So the first two steps are endoscopy with biopsy samples on which a microbiological diagnosis will also have to be made to know, for example, but not only if there is a prevalence of HER 2+. The CT scan study is important to understand what stage of the disease we are faced with. After this staging we find the first crossroads, neoadjuvant or ab-front surgery. In case of neoadjuvant we may find ourselves at a conversion attempt due to the presence of positive intercavoaortic lymph nodes, therefore the chemotherapy cycles will increase. However, in the case of perioneal carcinomatosis, studied with laparoscopy to also establish PCI, bidirectional therapy can be put into practice with the recent PIPAC. Even in this case it is possible to obtain a conversion and then undergo surgery. In all this, the parameters of our Japanese colleagues may be important and can represent an index to be taken into consideration for monitoring purposes. Also because we must not forget that the conclusion of these therapies is often the adjuvant treatment (DOI: 10.1097/CAD.0000000000000877 to be cited in the bibliography) which completes the treatment and where the aforementioned indices still have their precise role.
Comments on the Quality of English Languagegood english
Author Response
Dear Reviewer,
We would like to thank you for the suggestions and comments. All revisions and improvements are highlighted yellow in the revised version of our manuscript. We would like to thank you again for your valuable time and comments for strengthen our paper.
Best regards,
Reviewer 2 Report
Comments and Suggestions for Authors
The authors present an original research focusing on the the role of two biomarkers CEA / Albumin and CRP / Albumin ratio, in predicting the outcomes of neoadjuvant chemotherapy in advanced gastric cancer. The article is written in a concise and clear manner. The Introduction give synthetic information regarding the current protocol for neoadjuvant chemotherapy in gastric cancer. The methodology is clearly described and replicable. The results are based on statistic analysis of the data.
The discussions could be enriched, with more data regarding how these 2 biomarkers could be correlated with the response after chemotherapy and what values were obtained by other previous studies. A paragraph with study limitations should be added.
Author Response
Dear Reviewers,
We would like to thank you for the suggestions and comments. All revisions and improvements are highlighted yellow in the revised version of our manuscript. We would like to thank you again for your valuable time and comments for strengthen our paper.
Best regards,